# PeerJ

# The impact of social context on learning and cognitive demands for interactive virtual human simulations

Rebecca Lyons[1], Teresa R. Johnson[2], Mohammed K. Khalil[2] and Juan C. Cendán[2]

[1] Department of Psychology, Institute for Simulation and Training, University of Central Florida, Orlando, FL, USA
[2] University of Central Florida College of Medicine, Health Sciences Campus at Lake Nona, Lake Nona, FL, USA

## ABSTRACT

Interactive virtual human (IVH) simulations offer a novel method for training skills involving person-to-person interactions. This article examines the effectiveness of an IVH simulation for teaching medical students to assess rare cranial nerve abnormalities in both individual and small-group learning contexts. Individual ($n = 26$) and small-group ($n = 30$) interaction with the IVH system was manipulated to examine the influence on learning, learner engagement, perceived cognitive demands of the learning task, and instructional efficiency. Results suggested the IVH activity was an equally effective and engaging instructional tool in both learning structures, despite learners in the group learning contexts having to share hands-on access to the simulation interface. Participants in both conditions demonstrated a significant increase in declarative knowledge post-training. Operation of the IVH simulation technology imposed moderate cognitive demand but did not exceed the demands of the task content or appear to impede learning.

## INTRODUCTION

With recent advances in technology and virtual reality, *interactive virtual humans* (IVHs) are poised to revolutionize the training of skills involving person-to-person interactions. Demand for training in areas such as leadership, interviewing, cultural awareness, and negotiation is widespread and mastery of such skills is critical to performance in many fields (e.g., medicine, military, and customer service). Initial reports on the use of IVHs for training appear positive (*Babu et al., 2011*; *Kenny, Parsons & Rizzo, 2009*; *Kenny et al., 2009*; *Parsons et al., 2008*), but it is not yet known how interfacing with virtual humans influences the learning process. To extend our understanding of IVHs for learning, we explore the cognitive demands of operating an IVH system within the context of a complex problem-solving task and determine if medical students are able to effectively learn targeted medical content by using IVHs to practice patient interviewing and diagnosis. Furthermore, as group learning is increasingly integrated within educational curricula (*Johnson & Johnson, 2009*), we explore if the traditional benefits observed in group learning

Corresponding author
Juan C. Cendán,
Juan.Cendan@ucf.edu

translate to IVH-based training activities. Answering these questions has important practical implications for the future development and implementation of IVH in both academic and organizational learning contexts.

## Interactive virtual humans

IVH training represents a new era in the training of complex interpersonal skills. Prior to IVH simulations, experiential learning opportunities targeting the development of skills requiring person-to-person interaction were restricted primarily to peer-to-peer role-play, video demonstration, or on-the-job training exposure. In many prominent domains, IVH agents are filling the roles once held by human counterparts; however, implementation of IVH training systems has been most visible in the military (e.g., *Babu et al., 2011*; *Culhane et al., 2012*; *Hill et al., 2003*) and healthcare (*Cook, Erwin & Triola, 2010*; *Ziv, Small & Wolpe, 2000*; *Ellaway et al., 2009*). A variety of systems are now available that integrate IVHs to teach skills including non-native cultural conversational verbal and non-verbal protocols (*Babu et al., 2011*), cultural awareness (*Camacho, 2009*), customer service training, interviewing (*Camburn, Gunther-Mohr & Lessler, 1999*; *Hubal & Frank, 2001*), tactical questioning (*Department of the Army, 2006*), negotiation, leadership skills (e.g., *Rickel et al., 2002*; *Hill et al., 2003*), and clinical interview training (*Parsons et al., 2008*; *Kenny, Parsons & Rizzo, 2009*).

There has been rapid movement to integrate IVH practice opportunities into the training curriculum. Though work in this elaborate simulation format is relatively nascent, there appears to be a lag in research initiatives compared to the rate of application. Publications have emerged predominantly from the computer-sciences, artificial intelligence, and modeling and simulation communities and concentrate on understanding how best to realistically model human physical appearance, body motion (e.g., behavioral gestures, facial expressions, visual gaze), communication capabilities (i.e., agent perception and interpretation of communicative information from a human-user, agent provision of an appropriate and logical response), and the expression of emotion (*Kenny et al., 2007*; *Stevens et al., 2006*; *Swartout et al., 2006*). Simultaneous efforts to understand the influence of specific virtual human appearance and functionality, variations in user acceptance, and perceived realism of a virtual human interaction have also been undertaken (e.g., *Stevens et al., 2006*). This research has cumulatively contributed to the enhanced functionality and realism of IVHs, allowing for lifelike interactions between humans and these virtual counterparts.

Despite substantial progress in virtual human modeling, essential questions have yet to be examined regarding how the implementation of such simulation activities may influence the learning process. One essential consideration in shifting to IVH training systems is the level of cognitive demand required to effectively operate and learn within such training environments. In the following sections we describe the role of cognitive demand in learning and the rationale for incorporating cognitive demand as a central construct in research involving novel instructional formats, such as IVH-based training interventions.

## Cognitive load and learning

Cognitive load theory (*Sweller, 1988*; *Sweller & Chandler, 1991*; *Sweller & Chandler, 1994*; *Paas et al., 2003*) suggests that to optimize the instructional design of a learning system the design must accommodate the architecture of human cognition. Thus, an essential cognitive component to consider in the design process is working memory. As the primary memory system involved in processing new information and preparing it for storage in long term memory, as well as retrieval of stored knowledge, working memory is mandatory for learning; however, it has finite capacity and duration (*Baddeley, 1997*), which consequently limits the amount of information a learner can manage at any given time. Learning will be diminished if the cognitive resources required to process targeted learning content exceed those available. Instructional designers must be cognizant of the cognitive demands of a learning intervention to ensure the instructional system does not burden the learner.

To help conceptualize the various sources in competition for cognitive resources during learning, *Sweller (1988)* distinguished three types of cognitive load: intrinsic, germane, and extraneous. *Intrinsic load* references the inherent complexity of the material to be learned. This load form increases as learners are required to simultaneously manage multiple information elements (*van Merriënboer & Sweller, 2005*). To translate the training content to knowledge, learners allocate cognitive resources to attend to and make sense of training content. Resources applied toward these processes are referred to as *germane load*. Germane load is essential to learning as these efforts ultimately enable schema construction and the storage of knowledge in long-term memory (*Beers et al., 2008*). The third load type, *extraneous load*, comes from factors external or irrelevant to the content to be learned. Though there are infinite potential sources that may compete for one's attention during learning, one pertinent example is the cognitive demand associated with the technical operation of any media through which a training intervention is administered. Cognitive load of this form has the most power to detract from learning by consuming cognitive resources that could otherwise be allocated for learning (*Khalil et al., 2005*). Effective learning requires that the cumulative load from these three sources not exceed available memory resources (*Paas et al., 2003*).

In accordance with cognitive load theory, instructional methods should limit extraneous load in order to increase cognitive resource availability for intrinsic and germane load demands (*Khalil et al., 2005*). As most training interventions involving IVH target high complexity tasks, particular attention must be paid to minimize sources of extraneous load. If operation of an IVH simulation depletes memory resources before learning content can even be accessed, remaining resources may be insufficient to achieve the desired learning objectives. Responsibility falls to instructional designers to ensure the cognitive demands imposed by the technical or structural design of a learning task do not overburden learners.

Many IVH interfaces are operated using a traditional mouse and keyboard. Though most learners (particularly in upper level education) are well versed with these tools, they are used to perform functions in the virtual environment that they would not be used for

in real-world performance. For example, a physician obtaining history information from a patient would typically talk with the individual using spoken dialog; whereas many IVH systems elect to utilize chat-based communication system where messages are typed to the virtual human. The added requirement of typing to communicate is often reported by learners to break the natural flow of conversation and disrupt train of thought. IVH systems must be designed with great concern for ease of use and to be minimally demanding.

While designing IVH systems to require minimal cognitive demand is essential to ensure the opportunity for learning, from a practical standpoint, it is equally important to consider the *instructional efficiency* of the system. Instructional efficiency as defined by (*Paas & van Merriënboer, 1993*) is the mental effort required to perform a task in relation to the quality or accuracy of task performance. Efficient instructional systems produce learning benefits with modest cognitive load for maximal learning gain. A concern, particularly with a new training system, is that the effort required to operate the technology will detract from learning. Estimation of the cognitive demands and instructional efficiency of IVH systems have not yet been reported in the literature. To gauge the current design effectiveness of such systems it is critical to assess not only the cognitive demand an IVH modeling system places on human learners, but also the efficiency with which cognitive demands translate to observed learning.

## Small group learning

In many educational settings, particularly within higher education, emphasis is placed on the pursuit of individual knowledge. Yet despite individual accountability for learning, in the process of learning, small group learning is increasingly utilized in higher-level educational settings and has shown to be effective within a number of contexts for enhance learning outcomes (*Springer, Stanne & Donovan, 1999*). In terms of IVH-based training, arguments can be made both for and against small group learning.

Collaborative learning structures are touted for their capacity to enhance learning outcomes such as learning effectiveness, efficiency, and engagement, beyond those observed for individual learners (e.g., *Gokhale, 1995*; *Klein & Doran, 1999*; *Lou, Abrami & d'Apollonia, 2001*; *Nieder et al., 2005*). Synthesizing the research on group learning, Lou and colleagues (*2001*) report meta-analytic evidence that students benefit more from small group versus individual learning opportunities. Similarly a meta-analysis of group learning in undergraduate science, mathematics, engineering, and technology courses also found benefits to group learning, reporting enhanced achievement, greater persistence for learning, and more positive attitudes (*Springer, Stanne & Donovan, 1999*).

It is largely accepted that such benefits are attributable to the interactions that take place between group members. The collaborative behaviors in which groups engage facilitate a deeper level of information processing (i.e., the integration of new content into existing knowledge structures) and promote the future adaptability of learned knowledge skills and abilities to novel scenarios (*Kraiger, 2008*). Additional benefits observed in small groups have been credited to the diversity of knowledge, perspectives, and previous experiences group members contribute to group discussion. Group members simultaneously teach and

learn from one another. Each member contributes based on personal strengths and in areas of personal weakness, benefits from others' knowledge. Thus, group learning structures offer a form of built in support for learners.

Though group learning has generally proved an effective method for individual knowledge development, research examining the effectiveness of group use of computer-based simulations for learning reports mixed findings (*Lou, Abrami & d'Apollonia, 2001*; *Schlecter, 1990*). It should not be assumed that in all cases group learning will be more effective than individual learning (*Johnson & Johnson, 2009*; *Slavin, 1996*). Consideration must be given to the unique features of IVH training interventions and the influence they may have on learning in groups.

Commonly noted benefits of IVH systems may be mitigated in group learning settings. One factor that may be adversely affected is learner engagement. Unlike paper-based problem solving tasks or performance tasks in which all team members can simultaneously participate, computer-based systems are generally operated by a single set of controls (e.g., mouse and keyboard). Hence, only one human operator can be literally 'hands on' at a time. Due to the person-to-person focused knowledge and skills targeted by IVH instructional systems, the individual controlling the computer interface is the only learner to directly engage the IVH. Being in the operator position may make the difference between a learner feeling engaged in the interaction versus a third party spectator. As learner engagement is a central to learning effectiveness (*Noe, Tews & McConnell Dachner, 2010*), it is necessary to determine if engagement is suppressed by the constraints of IVH in group learning contexts.

Individual control over the pace of learning is another instructional feature influenced in a group learning contexts. Computer-based learning systems often afford learners a greater capacity to control the progression of learning. More time can be spent on topics a learner finds challenging and a learner may move quickly through familiar or easy content. Comparing self-paced and fixed-rate learning outcomes, *Tullis & Benjamin (2011)* demonstrated that self-paced learners outperformed those with structured training schedules, despite equivalent study time across these conditions. Thus, IVH-based learning systems are likely to force quick learners to pause; meanwhile, others may be left behind if their group chooses to forge ahead before they are ready. Examining how learners experience IVH systems in individual- versus group-learning structures is essential to understanding how IVH-based training interventions can most effectively be implemented within academic environments.

## The present study

The research presented in this paper investigates the effectiveness of an IVH-based training system in higher education. There were three objectives for the present investigation. First, this literature adds to the much needed validation research by demonstrating the validity of IVH-based training in higher education. The second objective for this study was to assess the cognitive load and instructional efficiency related to learning from an IVH-based instructional system. To understand the cognitive demand required by this new complex

simulation modality, we measured participants' perceptions of the total cognitive load associated with the learning task, as well as the specific cognitive demands of learning the task content (i.e., germane cognitive load) and operating the simulation technology (i.e., extraneous load). Based on cognitive load theory we expected that learners would perceive the technological demands of the IVH system to be low to moderate. This would indicate that cognitive resources are maximally reserved for comprehension of the learning content. Intrinsic cognitive load (i.e., the complexity of the material to be learned based on learner expertise) was not directly measured, but the simulation scenarios were predicted to pose moderate to high intrinsic cognitive load. The load estimation was based on the relatively novice experience-level of participants and the complexity involved in simultaneously gathering and processing information related to the patient's medical history and physical exam findings while formulating a diagnosis.

Implementation of novel instructional methods requires not only an understanding of the training system, but also consideration for how the training will be most effectively implemented. One decision educators must make is whether to use group social-learning structures. Thus, the third objective of this research was to examine the effect of social context of learning on learning outcomes. We manipulated the social context of IVH training (i.e., IVH system use in individual versus small group format) to examine the influence on the outcomes of learning effectiveness, the perceived cognitive demand of the learning task, instructional efficiency, and learner engagement. A combination of quantitative and qualitative methods were used to provide a more comprehensive understanding of participants experience with the IVH system.

## METHOD

### Participants and design

Fifty-six second-year medical students (54% female) participated in this research, with a mean age of 24.7 years (range = 21 to 37). All participants had previously received basic neuro-anatomy training within a course completed five-months prior to the present simulation activity. Pre-training survey data indicated that on average, participants reported moderate confidence in their knowledge of neurology and in their ability to correctly diagnose a patient with a neurologic condition. Only seven percent of students reported prior use of any neurologic simulator, and 78% considered themselves to have only novice to moderate prior experience in playing first-person perspective PC games.

This study utilizes a 2 (knowledge pre vs. post training) $\times$ 2 (individual vs. group study) mixed factorial design. Participants were randomly assigned to complete the simulation activity in one of two instructional conditions: (a) as an individual working alone at a computer ($n = 26$) or (b) as a member of a three-person group ($n = 30$) working collaboratively, face-to-face, on a shared computer. Students in the group study condition where then again randomly assigned to work in 1 of 10 three-person study groups.

Approval for this study was granted by the University of Central Florida Institutional Review Board (IRB approval number SBE-11-07533). Informed consent was verbally obtained, as well as notated via an online survey item, prior to participation.

## Procedure

The simulation activity was scheduled as part of a regular course simulation session; however, participation in the research was voluntary and had no influence on course performance. Prior to the simulation activity all participants, regardless of instructional condition, independently completed a brief demographics questionnaire and a knowledge test. Participants then separated into their respective treatment conditions for the simulation activity. In the group condition, participants were instructed they would work with their group members on a shared computer, and that they should collaborate with one another by sharing information and discussing the problem task.

The official educational simulation session began with a 10-min comprehensive tutorial on the Neurological Exam Rehearsal Virtual Environment (NERVE) platform, the simulation platform through which the training content was presented. NERVE is a complex, PC based IVH simulation platform designed to provide medical students with exposure to virtual patients presenting with potential neurological conditions. In NERVE, virtual patients are contextualized within a virtual examination room and all present with impaired vision (e.g., double or blurry vision). Students are then able to interview the patient using an interactive chat function and to perform several portions of the physical exam related to cranial nerve assessment (see Fig. 1): (a) ophthalmoscope exam, (b) visual acuity test, and (c) ocular motility test. The virtual patient is able to follow a number of behavioral instructions the student may request in formulating a clinical diagnosis (e.g., cover your left eye, look straight ahead, and stick out your tongue).

Following the system tutorial, participants completed three distinct clinical cases. The cases were presented in order of increasing complexity: (1) a condition with similarities to a cranial nerve condition, but that was not cranial nerve related; (2) an abnormality affecting the third cranial nerve (CN III); (3) an abnormality affecting the sixth cranial nerve (CN VI). All three virtual patients similarly reported experiencing visual impairment, but only the latter two cases were attributable to cranial nerve-related conditions. The non-cranial nerve case was included to serve as a reference of normalcy for students, and also because clinical diagnosis requires both the ability to positively identify when a condition is present, as well as the capacity to appropriately rule out alternative diagnoses. Prior to seeing the first virtual patient the students were told only that they were to assess the patients for potential cranial nerve abnormalities.

As part of each clinical case, students were asked to independently submit case notes related to the simulated patient's condition, as well as a suspected diagnosis. This form was submitted independently by each student. Those in the group condition were explicitly encouraged to work together and discuss their responses with group members prior to submission; however, the final response of each group member was left to his or her discretion.

Each case was followed with a detailed Flash® based case analysis, pre-generated by the course instructor (author JC), intended to serve as a form of feedback. These case analyses provided the correct patient diagnosis, as well as a description of the reasoning process the instructor used to reach the diagnostic conclusion. The feedback clearly indicated

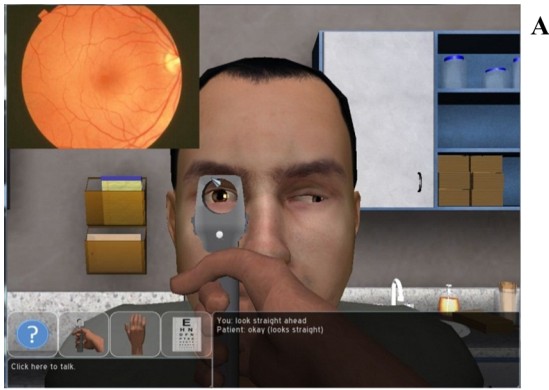

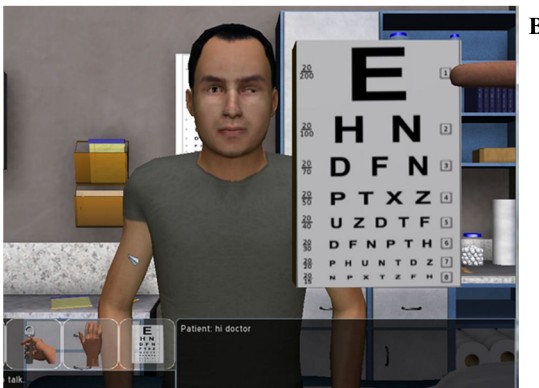

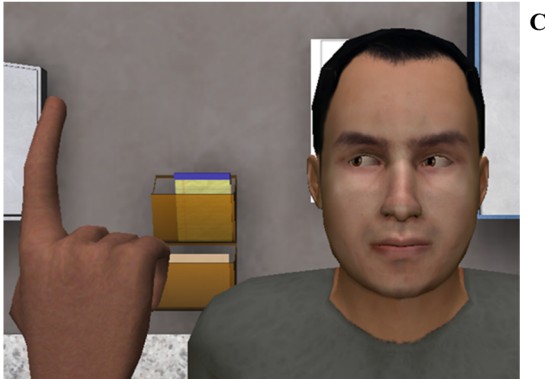

**Figure 1 Demonstration of the three visual tests that can be performed within NERVE.** The ophthalmoscope exam (A) is used to check for abnormalities in the nerves at the back of the eye. The visual acuity test (B) is used to check for vision problems. The ocular motility test (C) looks at eye movement and range of motion, as the virtual patient follows the virtual finger with his gaze.

which physical exam and history components supported the diagnosis (including pictures and diagrams depicting the physical exam and phrases from the patient history) or excluded other diagnoses. Thus, the case analysis provided students not only with the correct answer but also the logical steps to obtain a correct diagnosis for a patient with similar symptoms/history. Students were expected to self-evaluate their performance based on the expert case analysis. Case analysis content was identical for all participants. To

mitigate the potential influence of time pressure on cognitive load ratings (see *Paas & van Merriënboer, 1993*), sufficient time was allotted in the session agenda to allow students to work at their own pace without imposing time pressure. Progression through the three clinical scenarios was self-paced; however, on average, participants in both conditions spent 100 min completing this portion of the activity. Thus, the time spent on the tutorial and learning activity was comparable for both individuals and groups.

Following the learning session, participants were asked to independently complete a brief survey regarding their experience with the NERVE simulation activity and a post-knowledge test was administered.

## Measures

### Engagement

Participant engagement in interacting with the virtual patient was assessed using a single item. Participants rated their agreement with the statement "The simulation scenarios held my attention" using a 5-point Likert scale ranging from 1 = *strongly disagree* to 5 = *strongly agree*.

### Knowledge and learning

A 12-item test consisting of seven multiple choice items and five fill-in-the-blank items was used to assess declarative knowledge related to the assessment of clinical pathology. All questions consisted of a brief descriptive statement about a patient from which the student may infer the most probable site of anatomic dysfunction. For example, one item read:

> A four year old girl comes to your office for a routine physical. Her examination is normal except her left pupil is nonreactive. When you shine a light in her right eye, the right pupil constricts. When you shine a light in her left eye, the right pupil constricts. Where is the most probable site of anatomic dysfunction?

Some items also included additional visual information, with either an eye position chart (see Fig. 2) or a brief video (15 s) of the patient's eye exam. Participants received 1 point per item for identifying the correct anatomic dysfunction. To capture learning, this knowledge test was administered to participants pre- and post-simulation. Learning was then calculated as the difference score of pre- and post- test performance.

### Cognitive load

The cognitive load of the NERVE simulation activity was measured based on a modified *Paas & van Merriënboer (1993)* scale. Participants were asked to rate the amount of effort they utilized for the following three items: (a) the learning event as a whole, (b) learning the medical content presented in the experience, and (c) using the new simulation technology. Ratings were based on a 5-point Likert scale ranging from 1 = *very low mental effort* to 5 = *very high mental effort*.

### Instructional efficiency

Instructional efficiency was calculated based on the equation proposed by *Paas & van Merriënboer (1993)*. In context of the present research, efficiency reflects the total cognitive

**Peer**J

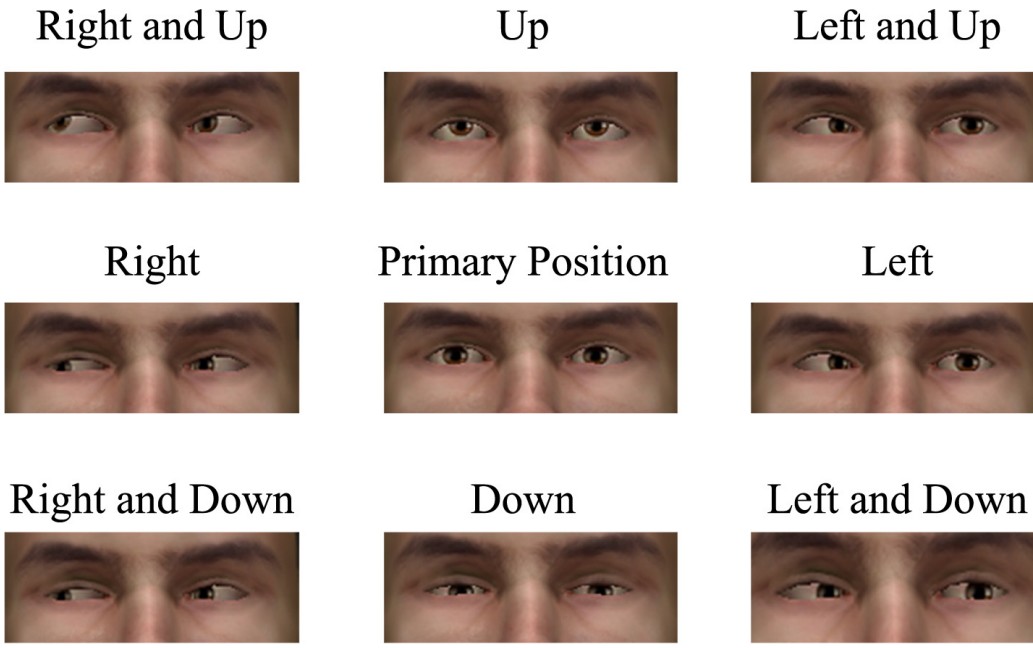

**Figure 2 Example eye position chart demonstrating each of the nine eye positions examined when performing an ocular motility test.** The test conductor observes for abnormal eye movement. As a declarative knowledge test item, the students check the images to determine if a cranial nerve abnormality may be present. In this picture sequence, in the 3 images on the right hand side (where the patient is looking to the left), we see the left eye remains relatively straightforward, rather than shifting away from the nose as would be expected.

load required to learn to assess and diagnose virtual patients presenting with probable cranial nerve abnormalities in light of the mental effort associated with learning through use of the NERVE educational technology. Mathematically, efficiency was calculated using mean standardized learning ($z$ Learning), reflected by improvement in clinical pathology declarative knowledge, and the standardized mean overall mental effort associated with the learning activity ($z$ Effort):

$$\text{Effciency} = \frac{z \text{ Learning} - z \text{ Effort}}{\sqrt{2}}.$$

## Data analysis

All data in this study were collected and analyzed at the individual-level as the purpose of this study was to understand how the social nature of the learning context changes the educational experience at the level of the individual learner. Also, individual knowledge and skill development are the criteria by which success in medical school is ultimately dependent. Thus, an examination of data at the group-level, as a unit of analysis, was not relevant for the present research.

Prior to conducting statistical analyses, the data was examined to ensure assumptions of relevant statistical tests were met. To compare learning across the study conditions,

**Table 1 Pre- post- knowledge test scores for individuals and groups.**

| Condition | N | Pre-test M (SD) | Post-test M (SD) |
|---|---|---|---|
| Individual | 26 | 7.65 (2.19) | 8.96 (2.34) |
| Group | 30 | 8.00 (2.20) | 9.97 (1.30) |

**Notes.**
The mean score reflect the number of correct items out of 12 possible items.

a 2 (knowledge pre vs. post training) ×2 (individual vs. group study) mixed factorial ANOVA was conducted. Mann–Whitney U-tests were used to test for differences in the medians [minimum, maximum] of individual and group engagement and cognitive load. Responses to open-ended survey items were qualitatively examined via a thematic content analysis (conducted by author RL). This analysis technique was used to identify the most common themes in students' comments for what was liked most and least about the IVH activity, as well as how working in an individual or group setting influenced their experience.

# RESULTS

## Learning

Across both conditions, students demonstrated significant knowledge gains from pre-to post-test, $F(1, 54) = 26.84$, $p < 0.001$, $\eta_p^2 = 0.33$. The interaction term for learning and social context of learning was not statistically significant; although, practically, students in groups performed nearly one (out of 12) items better on the post-test. See Table 1 for pre-post knowledge score means by condition.

## Engagement

Participants in the group and individual practice conditions, on average, did not significantly differ in their ratings of perceived level of engagement in the IVH learning activity ($p = .58$; median rating = 4 [2,5] and 4[1,5], respectively).

## Cognitive load

As a whole, participants reported the overall cognitive effort required by the NERVE—IVH task to be moderate (median rating = 4[1,5]). Overall mental effort as reported by individuals and groups indicated perceptions were not significantly different across conditions ($p = .11$; median rating = 3 [1,5] and 3[1,5], respectively). Of the three mental effort ratings, technology effort was rated the lowest by both individuals (median rating = 2 [1,4]) and groups (median rating = 3 [1,4]). The difference between individual and group technology mental effort was not significant ($p = .28$). The comparison of individual and group content effort ratings also was not statistically significant ($p = .13$; median rating = 3 [1,4] and 3 [2,5], respectively). See Fig. 3 for a side-by-side comparison of these cognitive load ratings by condition.

To further examine the relationship between the various ratings of cognitive load a bivariate correlation analysis was performed for each instructional condition (see Table 2).

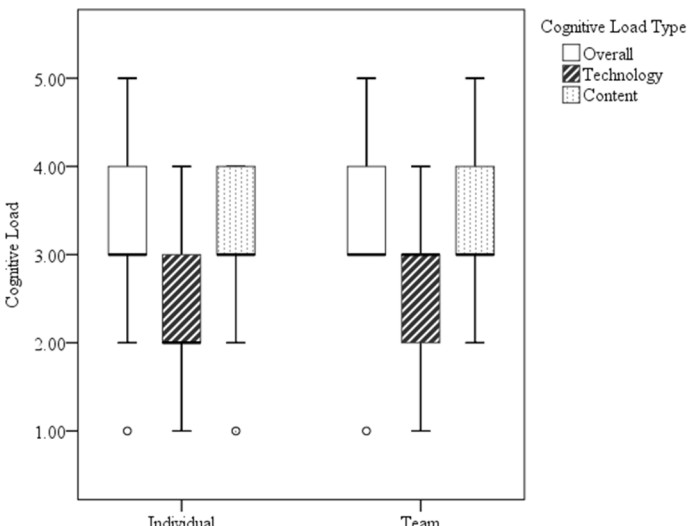

**Figure 3 Boxplot of individual and group cognitive load ratings.** Boxplot of technology specific, content specific, and overall cognitive load ratings of participants in the individual and group learning conditions. Whiskers represent the minimum and maximum rating values for each boxplot.

**Table 2  Intercorrelations among cognitive load forms per instructional condition.**

| Variable | 1 | 2 | 3 |
|---|---|---|---|
| 1. Overall | — | .14 | .17 |
| 2. Content | .38[*] | — | .24 |
| 3. Technology | −.31 | −.29 | — |

**Notes.**

Correlations for the individual learning condition ($n = 26$) are presented below the diagonal, and correlations for the group learning condition ($n = 30$) are presented below the diagonal.

[*] $p < .05$.

For the individual learning condition only, results indicated a significant correlation between users' overall mental effort rating and the perceived mental effort required by the scenario content, $r = 0.38$, $p < 0.05$. Perceived mental effort required by technology use did not correlate significantly with either overall mental effort or content mental effort for either condition.

## Instructional efficiency

The $z$-score values for overall mental effort and performance involved in these calculations are presented in Table 3.

The instructional efficiency was relatively similar for individual ($E = 0.03$) and group ($E = −0.03$) learners (see Fig. 4). The graphic suggests that performance was slightly better for those utilizing the NERVE—IVH system in groups versus those working as individuals. Mental effort also trended towards being greater for those in the group condition.

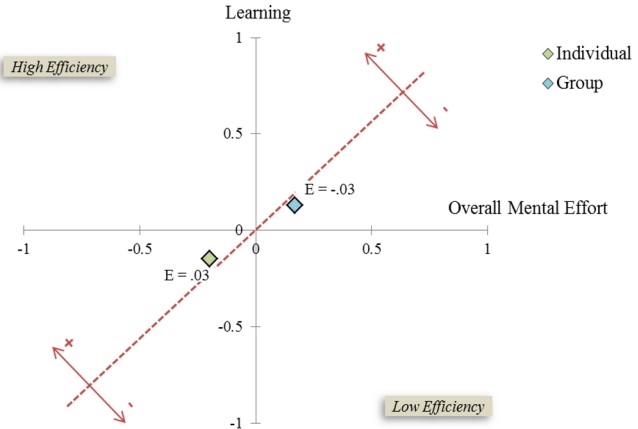

**Figure 4 Instructional efficiency graph for the training system as experienced by individuals and groups.** The *x*-axis represents mean overall mental effort. The *y*-axis represents performance. The gray dotted line reflects the midline for efficiency, such that efficiency scores to the left of this line indicate high instructional efficiency and scores to the right of this line indicate low instructional efficiency.

**Table 3 Z-score values used for calculating instructional efficiency.**

| Condition | N | Overall mental effort | Learning |
|---|---|---|---|
| Individual | 26 | −0.20 | −0.15 |
| Group | 30 | 0.17 | 0.13 |

## Qualitative analysis of user comments

### IVH system relevant comments

Students' written comments revealed a variety of factors they liked about the IVH simulation experience. The most commonly praised feature was that the system allowed access to rare clinical conditions beyond those seen in clinical rotations or other simulation modalities. This was noted by 16% of students. Also, 5 or more students each commented positively on the interactive nature of the simulation, the provision of immediate feedback, the perceived educational benefit of the IVH activity, the ease of use, the ability to communicate with the virtual patient, and the comprehensive nature of the experience.

When asked what they would most like to see improved about the system 85% of students indicated issues related to communicating with the virtual patient. The most common complaint, mentioned by 60% of students, was that the IVH patient either misunderstood or was unable to answer questions to which the student would have liked information. Additional comments related to a preference for communicating via verbal dialog versus typing or a desire that the virtual patient responded with greater detail.

### Individual versus group benefits

Of the 26 participants in the individual learning condition, 50% expressed the ability to work at one's own pace as a benefit to working independently. An additional 35% felt they were more mentally challenged by working independently and that the absence of

others to fall back on when challenged promoted critical thinking. When asked about the perceived disadvantages of working as an individual, 50% of those in the individual study condition felt that working with another student would have helped them to conduct a more comprehensive patient interview and/or neurological exam. Similar comments expressed the benefits of peers as a sounding board for ideas and diagnosis (27%) and ability to ask questions of peers (11%).

From the perspective of the 30 participants in the group learning condition, 86% reported the primary advantage to working in groups was that it helped by providing ideas and alternative perspectives, allowing them to ask questions, and "fill[ing] in each other's gaps in knowledge". Three themes were identified as negatives by those in the group learning condition. The most commonly cited criticism of group study, noted by 34% of group participants, was the lack of control over the pacing of the exercise. Most of these individuals reported feeling the group slowed the process. Furthermore, though all students had the opportunity to control the IVH for at least one patient case, 17% noted they would have preferred more hands-on control of the IVH system. Finally, 17% of the respondents felt they would have preferred to test their individual knowledge by working alone, as they would have to do in real life.

## DISCUSSION

Implementation of novel instructional methods requires an understanding of a system's effectiveness for learning, as well as how a given system can be optimally deployed. One objective of this study was to validate the effectiveness of an IVH-based training system for training the diagnosis of neurologic abnormalities. Utilization of IVHs within individual and small group learning contexts was examined. To our knowledge this is the first study to consider the cognitive load and instructional efficiency of an IVH-based training system.

### Is IVH effective for learning complex problem-solving tasks?

The instructional content targeted by the NERVE system addressed the complex problem-solving task of diagnosing cranial nerve abnormalities. In this task students practiced both patient interviewing and examination skills, and used this information to formulate a suspected diagnosis. In this learning context, the IVH system was found to be an effective instructional tool for use in both individual and small group learning contexts. Learners were able to engage the IVH both through conversation and physical examination to obtain the information required to infer a medical diagnosis.

The structure of the learning activity supports the conclusion that the observed learning can be attributed to engagement with the IVH activity versus some other source (e.g., repeated test exposure or directed perception). For example, though possible that some learning occurred due to repeated items on the pre- and post-test, students did not receive any feedback following the pre-test (i.e, neither test scores nor item answers) and progressed directly from the pre-test to the IVH activity, and then continued immediately to the post-test. This suggests that the learning took place as a result of engagement and reflection within the IVH activity versus the repeated exposure to the test items. It is equally unlikely that learning outcomes were influenced by directed perception due

to the cased-based nature of the test questions and IVH activity. Specifically, because the patient scenarios and details of the physical presentation of any given cranial nerve abnormality differenced in the IVH learning activity and the test items, the similarities between the cases in the learning and test contexts were not immediately salient. Also, the primary purpose of the course simulation session within which the IVH learning activity was embedded was for the students to gain practical exposure to assessing cranial nerve abnormalities. The students were not told they would be given a post-test following the learning activity and were not graded on either test. Thus, extrinsic motivators to focus on learning the specific test answers were minimized.

The primary threat to the external validity of the study findings would be the use of a single, specialized educational population (i.e., medical students) who are adept at the use of computers. Thus, generalizations from this study to less computer adept populations may not translate. Future research is required to examine the boundaries of tasks for which IVH systems can effectively promote learning. For example, in NERVE, unless engaged, the virtual human remains largely inactive. Future research should validate systems in which the virtual human plays a driving role in the interaction (i.e., inserting actions that must be observed by the learner without learner prompting).

## How cognitively demanding is operation of the IVH system?

When intrinsic load is high, simulation developers strive to generate simulation systems that mitigate the cognitive demands placed on the learner. As it is known that the learning tasks addressed by IVH-based training are complex, an objective of this research was to examine the extent to which operating the NERVE training would add cognitive load to the learning activity. Self-report data of the cognitive load required to operate the IVH technology indicated modest cognitive demands. Also, the reported cognitive demands of training content exceeded those of the technology, as would be desirable for effective learning. This helps to ensure learners are allocating working memory resources to the learning of task content rather than the operation of the training technology. Though no formal guideline has been established within the literature regarding the degree to which content cognitive demands should exceed technology cognitive demands and because technology cognitive load describes the cognitive demand of learning to operate and operating the training technology (i.e., technology that is not part of the task/material to be learned) and that the three types of cognitive load are assumed to be additive, we can reasonably speculate that it would always be desirable to minimize the technology cognitive load. We equally speculate there is a point of diminishing returns where it is of limited benefit to further attempt to reduce technology cognitive load; as long as sufficient cognitive resources are available to accommodate the content cognitive load and intrinsic cognitive load.

In considering the reported cognitive load due to technology—in relation to student comments reflecting near unanimous agreement that they would like to see this feature of the training improved–it is somewhat surprising that individuals did not perceive a greater cognitive load. This suggests that although there is room for improvement on the communication feature of the NERVE training system, the inconvenience was not such

that it impaired students' ability to learn from this system. It is also possible that results from the thematic content analysis were biased due to the use of a single-rater in this process; however, this risk is limited due to the straightforward nature of the questions and responses analyzed.

Another interesting observation was that technology cognitive load was not significantly correlated with overall cognitive load perceptions for either study condition. Based on cognitive load theory, we would expect to see a correlation between both technology load and content load with overall cognitive load, given that these ratings are theoretically both subcomponents of overall cognitive load. In contrast, the observed results suggest that when asked to rate overall cognitive difficulty, participants either did not consider, or placed less weight on cognitive demands of the technology. This observation is important particularly when gauging cognitive load of technologies, because if the difficulty of technology is not addressed explicitly, or if one only measures overall load, a highly demanding technology system may be overlooked. That is, it appears students conceptually consider primarily the content difficulty of training when reflecting on overall perceived difficulty, and may inherently perceive any difficulty experienced in relation to the training system itself as separate.

### Is group learning effective for IVH-based activities?

Research on small group learning has provided mixed evidence for its effectiveness, particularly related to computer-based exercises (*Klein & Doran, 1999*; *Lou, Abrami & d'Apollonia, 2001*). In the current study we tested the relative usefulness of an IVH-based activity when learning occurred either in an individual or small group study context. Overall, the results supported the IVH-based NERVE simulation activity to be an effective learning tool in both individual and small group study contexts. To mitigate potential speculation that learning in the group condition may be attributable to discussion of the pre-test items, groups were monitored via remote video feed during the IVH activity. In our observations, groups' discussion remained focused on the immediate IVH case scenarios. No references to the test items were heard. Though students in the group condition were encouraged to collaborate during the simulation activity, they were not allowed to interact with one another when completing either the pre- or post- tests. Furthermore, on average, group members did not score consistently with one another within groups ($sd = .00$–$2.51$); participant scores at post-test were only the same for 1 of the 10 groups, with an average within group $sd = 1.3$.

Because both those in the individual and group learning conditions increased their performance on the cranial nerve knowledge test, it can be inferred that the knowledge gain was due to the IVH activity versus something else. However, it was observed that variance in post-training performance scores in the team condition was a third of that observed in the individual condition post-test performance score, despite statistically equivalent performance scores and variance prior to training. This suggests that engaging with the virtual patient in a group setting has greater educational impact in that the group

provides an alternative method of instructional support. For individuals, the IVH training alone was effective for some, but not all.

It was also observed that despite concerns that learner engagement in IVH activities would suffer; this was not reflected in student self-reported engagement. If working in a group had any form of detrimental effect due to engagement with the system, it was counter-balanced by the added engagement of interacting with group members. Having said that, as groups are generally more engaging, and reported engagement was similar in both instructional conditions, there was likely some trade off.

Perceptions of cognitive load were also found to be similar across the study conditions. Learners generally reported moderate/high ratings of overall cognitive load and content cognitive load, and moderate technology cognitive load. Though statistically significant differences were not detected for any of the three cognitive load types, the comparisons of overall cognitive load and content cognitive load approached significance ($p = .11$ and $p = .13$, respectively) in favor of learners experiencing greater cognitive load when working in groups. It is possible the study sample size was insufficient to detect existing differences between the study conditions. A larger sample size may provide clarity to the subtle trend suggested in the data of groups perceiving greater cognitive load. Although the overall instructional efficiency levels of group and individual instruction were similar, the trend of this data, as shown in the efficiency graphic (see Fig. 4), suggests that group members exert greater cognitive effort; but, these group members also benefit from this effort, as reflected by the tendency toward higher performance scores. Though this interpretation of the data is cautionary, based on the statistical findings that independently compare performance and cognitive demand of individual and group study, we can conclude that the IVH-based NERVE simulation was at least as effective for small group study as individual study. Given the spectrum of scenarios for which IVH simulations are proposed, the effectiveness of groups may not generalize to tasks less focused on problem solving and which are more dependent on observation or behavior recognition (e.g., teaching non-native cultural conversational verbal and non-verbal protocols as done by the military).

Prompted by user reactions to the natural-language user interface in this study, our research team has already begun work on a modified IVH communication system that will include a text-matching feature to make it easier for students to obtain desired information from a virtual patient. With this modification, if a virtual patient is unable to recognize a message communicated by the human user the system will predict up to three alternative statements perceived by the system as similar or related to the typed message. Human users may then select from the list of alternative statements, or indicate that none of the options reflect their statement. It is anticipated that feature will not only improve accuracy of the human – IVH conversation, but by presentation of alternative question statements, may also contribute as a pseudo team member, by presenting ideas of questions the system recognizes to be important.

### Funding

This work was funded in part by the National Institute of Health, grant R01 LM010813-01 (Neurological Exams Teaching & Evaluation Using Virtual Patients). The funders had no role in study design, data collection and analysis, decision to publish, or preparation of the manuscript.

### Grant Disclosures

The following grant information was disclosed by the authors:
National Institute of Health: R01LM010813-01.

### Competing Interests

The authors declare there are no competing interests.

### Author Contributions

- Rebecca Lyons conceived and designed the experiments, performed the experiments, analyzed the data, wrote the paper, prepared figures and/or tables, reviewed drafts of the paper.
- Teresa R. Johnson conceived and designed the experiments, performed the experiments, analyzed the data, wrote the paper, reviewed drafts of the paper.
- Mohammed K. Khalil conceived and designed the experiments, wrote the paper, reviewed drafts of the paper.
- Juan C. Cendán conceived and designed the experiments, performed the experiments, wrote the paper, reviewed drafts of the paper.

### Human Ethics

The following information was supplied relating to ethical approvals (i.e., approving body and any reference numbers):

University of Central Florida IRB: SBE-11-07533.

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
