# Peer review of "The impact of social context on learning and cognitive demands for interactive virtual human simulations"

_PeerJ, doi:10.7717/peerj.372_

## Round 0.1 · original submission · Major Revisions

The reviewers found much to like in this paper. In considering a revision, please pay particular attention to Reviewer 1's comments on the experimental design, statistical analysis, confounds, and other areas in need of clarification. Reviewer 2's comments in these areas are certainly worth considering as well. Revisions that address these concerns would certainly strengthen the contribution of this paper.

·

Basic reporting

Overall good basic reporting. However, I have some suggestions to the authors:
1) I did not find any information concerning demographic data that was collected (line 249). This could probably increase the quality of the paper.
2) It would be very informative to undestand how much time was needed for the training (as well as for the test cases). In order to draw conclusions on learning and cognitive load, I believe that the groups should be comparable in this aspect.
3) Were the groups assembled at random or not? Did the students in the groups know each other beforerhand?
4) Line 274, should it be "Paas et al (1993)?
5) A qualitative analysis was performed for identifying themes in the students comments about their experiences of the IVH activity. How was this analysis performed (line 298-300)?
6) Line 327, "slightly higher", shouldn't it be "slightly lower"?
7) I would suggest that a measure of distribution should be added to the bars in figure 3.
8) Reference in line 338 should maybe be the same as in line 280?

Experimental design

Experimental design seems to be good, but some correctable flaws seem to exist.
1) The simulation scenarios were designed to possess high intrinsic cognitive load (line 173-174). Was this asserted?
2) N=30 in the group condition. However, since the students worked together and also were encouraged to collaborate one might suspect that their answers were very similar (otherwise they probably didn't collaborate). This argument would lead to that N=10 in the knowledge test for the group condition.
3) T-test was used for statistical comparisons. Were the samples normal distributed? Engagement data was assessed using a scale possessing ordinal scale properties, how do the authors argue their use of t-test for this?

Validity of the findings

In general the study adds important knowledge to the literature in paritcular concerning the use of small group education with interactive virtual humans. However, I would be interested in some clarifications concerning:
1) If the same test was given twice of course one might argue that there is some learning by doing the test (as well as directed perception to the content of the questions during the tutorial. A larger problem is if the students in the groups were able to discuss the test with their friends in the group condition but not in the individual learning condition. If this occurred, I have difficulties seeing that the results on knowledge have much value. Is the conclusion that the IVH system is an effective instructional tool (line 391) based on this, or something else?
2) Task characteristics for determining cognitive load measures in Paas and Merrienboer’s paper include time as a factor. How is this viewed upon by the authors? Did the students have the same amount of time to complete the training and the test (see above)?
3) Although not statistically significant different, could there still be a difference in cognitive load between the instructional conditions (type 2 error)? In this study sample size would then be a limitation.
4) As far as I understand, the bivariate correlation analysis of cognitive load was performed for the whole material. Was also an analysis for each instructional condition performed? It could for example be hypothesized that the technology mental effort would be influenced by working in a group (letting someone else deal with technical difficulties).
5) I agree with the authors, that the cognitive demands of training content should exceed those of the technology (line 409). Do the authors have any suggestion about how large the ratio desirably should be?

Additional comments

In general interesting and novel study. Good introduction and much of the discussion is of good quality.

·

Basic reporting

The article is well written and organized. It contains an ample introduction that frames the question of research and provides several focus points of background knowledge necessary for readers to understand the research study. The research study is well explained and has appropriate figures and tables. The conclusions are provided and there is adequate discussion of what they likely mean according to the authors.

Overall, I thought this was an original idea that was of value to the field of medical education and virtual patient science. The NERVE application was quite ingenious and the study will serve to guide future research in this area.

Experimental design

The study appears to be well designed and is adequately powered with an appropriate study group. IRB is in place. The group may be too expert to have picked up a significant technology effect as a factor in cognitive load as the medical students seemed to readily discount issues using the technology and separate that load from the cognitive load regarding the diagnostic tasks. Nevertheless, the findings are scientifically valid and the approach is sound.

Validity of the findings

The findings appear to be valid and reasonable. Unexpected data is explained with possible alternative interpretations by the author. Study limitations are discussed frankly as are weaknesses in the IVH simulation. The discussion and opinions of the authors are reasonable deductions from the research data returns. In the end, this may not have been the best group in which to look for cognitive load effects of the technology, but the results that show that represent useful data for educators and simulation authors.

Additional comments

This is a novel, interesting and useful research study that I was pleased to read. It is a sound study and it employs a useful virtual patient training scenario. The comparison of group learning use versus individuals was something I hadn't considered but was an important question to ask. Students seemed to rely on each other as a pedagogical guide within the small group and did well, despite the limited opportunities for interaction with the simulator.

---

## Round 0.2 · Minor Revisions

Thank you for your revisions to this paper. Your responses thoughtful consideration of the reviewer comments. My suggestions for further minor revisions focus on the clarification of these comments and the inclusion of some detail from your response letter in the manuscript. Specifically:

1. Reviewer 1, basic reporting, comment 2- your response indicates that the time spent on the tutorial and the learning activity were comparable for both individuals and groups, but this comment is not included in the paper. Please add it.

2. Reviewer 1, basic reporting, comment 5: The addition of the methods for the qualitative methods is appreciated. Given that this description does not describe validation by a second researcher, I suggest adding a comment in the limitations discussing the possible limits of the analysis.

3. Reviewer 1, basic reporting, comment 7: Please revise the caption to the revised Figure 3 to indicate that the extremes are min and max values.

4. Reviewer 1, experimental design, comment 1: These changes are fine, but the revised manuscript lost a paragraph break before the sentence starting with "Implementation..."

5. Reviewer 1, experimental design, comment 2: The question of group consistency should be raised in the discussion.

6. Reviewer 1, validity, comments 1, 2, and 5: You have provided clear responses to the reviewers' concerns, without indicating how these issues are covered in the paper. Please add some of this content to the discussion.

7. Reviewer 2, comment 1: I was not able to find the discussion of the external validity limitations. Please ensure that it is included in the document.

---

## Round 0.3 · accepted · Accept

Thank you for your careful and prompt attention to the requests for revisions.